# Analysis of the Lipolytic Activity of Whole-Saliva and Site-Specific Secretions from the Oral Cavity of Healthy Adults

**DOI:** 10.3390/nu11010191

**Published:** 2019-01-18

**Authors:** Weng Yuen Willy Lai, Jocelyn Wei Min Chua, Saloni Gill, Iain A. Brownlee

**Affiliations:** 1Newcastle Research and Innovation Institute, Devan Nair Building, Singapore 600201, Singapore; 1601459@sit.singaporetech.edu.sg (W.Y.W.L.); Jocelyn_chua03@hotmail.com (J.W.M.C.); salonikdang@gmail.com (S.G.); 2CSIRO Nutrition & Health Program, SAHMRI Building, North Terrace, Adelaide, SA 5000, Australia

**Keywords:** saliva, oral lipase, fat digestion, lipolysis

## Abstract

It is currently unclear how the process of fat digestion occurs in the mouth of humans. This pilot study therefore aimed to quantify the levels of lipolytic activity at different sites of the mouth and in whole saliva. Samples of whole saliva and from 4 discrete sites in the oral cavity were collected from 42 healthy adult participants. All samples were analyzed for lipolytic activity using two different substrates (olive oil and the synthetic 1,2-o-dilauryl-rac-glycero-3-glutaric acid-(6’-methylresorufin) ester (DGGR)). Bland–Altman analyses suggested that the two assays gave divergent results, with 91% and 23% of site-specific and 40% and 26% of whole-saliva samples testing positive for lipolytic activity, respectively. Non-parametric multiple comparisons tests highlighted that median (IQR) of lipolytic activity (tested using the olive oil assay) of the samples from the parotid 20.7 (11.7–31.0) and sublingual 18.4 (10.6–47.2) sites were significantly higher than that of whole saliva 0.0 (0.0–35.7). In conclusion, lipolysis appears to occur in the oral cavity of a proportion of individuals. These findings give a preliminary indication that lipolytic agent activity in the oral cavity may be substrate-specific but do not discount that the enzyme is from sources other than oral secretions (e.g., microbes, gastric reflux).

## 1. Introduction

Behind carbohydrate, dietary fat intake provides the highest percentage of total energy consumption, ideally ensures the necessary provision of essential fatty acids, and supports the uptake of fat-soluble vitamins [1,2]. The largest proportion of dietary fat is consumed in the form of triglycerides, which must be digested before they are in a form that can be absorbed in the small intestine [3]. In healthy adults, this process appears to be predominantly driven by pancreatic lipases [4]. Pre-intestinal lipases may have important roles in triglyceride digestion in newborns, alongside lipases secreted in breast milk [5,6] and in hydrolyzing a greater proportion of dietary triglycerides in individuals with pancreatic exocrine insufficiency [7]. The sites of pre-intestinal lipase secretion that have previously been evidenced in some mammalian species to be the oral cavity and the gastric epithelium [8,9]. Even low amounts of lipolysis in the oral cavity may be particularly important as this could represents a means by which dietary fats are sensed [10], thereby providing a stimulus for appetite control and energy intake regulation [11]. Studies in cows and rats have consistently highlighted the secretion of a lingual lipase from the serous (Von Ebner’s) glands found on the dorsal surface of the tongue [12,13,14], with a previous study suggesting low levels of lipolytic activity in homogenates of serous tissues from the human tongue [15]. The classical view of the existence of a human lingual lipase [5,16] is not always supported by more recent observations. Voigt et al., (2014) noted that, although lingual lipase expression could not be detected in their participants, lipolytic activity could be measured in the human oral cavity [17], suggesting the oral presence of lipase. Previous studies have also suggested that lipolysis does occur in the oral cavity of humans [2,18,19].

Lipolysis in the oral cavity could occur as a result of enzymes of microbial or endogenous origin, either produced in the oral cavity or present as a result of refluxed gastric contents [20,21]. If these enzymes are secreted from specific sites within the oral cavity, the most likely sites of production are the lingual papillae or the major or minor salivary glands [22]. The authors hypothesize that such site-specific secretion would mean that lipolytic activity could be localized. As a result, this study aimed to test the lipolytic activity of secretions collected from different sites in the oral cavity of healthy adults, as assessed using spectrophotometric methods in 96-well plate assays.

## 2. Methods

### 2.1. Participant Recruitment and Sample Collection

Following ethical approval by Newcastle University Faculty of Science, Agriculture and Engineering Ethics committee (16-BRO-0048, approval date 9th September 2016) and collection of informed consent, a total of 42 healthy adult participants (age range 21 to 30 years, *n* = 30 female) were recruited for this project. Exclusion criteria were: current use of medications that could impact on salivary flow, current oral, respiratory or bloodborne infections or being previously diagnosed with a major long-term health issue. To standardize the saliva collection procedures, participants were requested to attend visits first thing in the morning (0830 to 1000) and to avoid eating, drinking, brushing teeth or using mouthwash and engaging in moderate or high-intensity physical activity for at least 2 hours before the visit. They were also requested to not consume any alcohol, caffeine, or nicotine in the 12 hours prior to the visit, as these factors might temporarily affect salivary flow. Visits were also scheduled more than 24 hours apart from participants’ most recent dental check-up to further reduce the potential for changes to habitual saliva production [23]. Upon arrival at the volunteer suite, participants were requested to fill out a lifestyle questionnaire to confirm eligibility and to collect demographic data.

Whole-saliva samples were collected first. Participants were asked to allow saliva to pool in their mouth for a few seconds prior to spitting into a 50 mL centrifuge tube, containing 50 mg of citrate crystals as a preservative. Subsequently, site-specific samples were collected by participants using sterile cotton swabs (EUROTUBO^®^, Amadora, Portugal) under researcher supervision. Samples were always collected in the same order: inside the left cheek (at the opening of the parotid gland), along the lower lip (minor saliva glands), at the dorsal surface of tongue (near to the proposed site of the lingual lipase production), and below the tongue (sublingual/submandibular gland sampling). The swab was held in the specific sites by the participant for at least two minutes, ensuring that the swab was saturated with saliva prior to carefully removing it from the mouth without touching other surfaces [24]. Between each sampling, participants rinsed their mouth with water for approximately one minute to try and ensure that the saliva collected was produced at that particular site and to minimize the potential for contamination from other saliva produced elsewhere. The swab tips were then placed in 1.5 mL microfuge tubes and capped tightly.

Following completion of sample collection, the microfuge tubes containing the swabs were filled with 0.5 mL of phosphate-buffered saline (PBS) and allowed to soak for at least 5 minutes. The tubes were then vortexed, and the swab tips wrung out to maximize the amount of saliva that ended up in the solution. The saliva samples were subsequently centrifuged to remove particulate and cellular debris before extracting the resulting supernatant. All supernatant samples were then stored in a −80 °C freezer prior to analysis for lipolytic activity.

### 2.2. Assessment of Lipolytic Activity

All reagents described below were purchased from Sigma-Aldrich (Singapore). Spectrophotometric assays based on two different substrates were carried out. The first assay was based on the loss of turbidity of an aqueous olive oil in PBS (0.04% olive oil v/v) emulsion in the presence of bile acids (0.35% sodium taurodeoxycholate) [25]. This method has more recently been developed for use in 96-well assays to test the potential for dietary factors to impact on the processes of fat digestion [26]. Due to the lack of a lingual lipase standard, activity was assessed against a standard curve developed using porcine pancreatic lipase and colipase. All solutions were pre-incubated at 37 °C Turbidity within the wells was measured at 405 nm using a microplate reader (Tecan Infinite^®^ 200 Pro Microplate Reader, Tecan Group Ltd. Männedorf, Switzerland) straight after mixing samples/standards and the substrate and again after 30 minutes incubation at 37 °C. Loss of turbidity was compared to a porcine pancreatic lipase standard (used in a ratio of 3:200 with a final lipase concentration of 500 μg/mL in the working solution), which was used here due to a lack of commercially available lingual lipase. Lipase activity data have been expressed as pancreatic lipase equivalents (in μg/mL).

The second assay was based on a commercially available synthetic substrate (1,2-o-dilauryl-rac-glycero-3-glutaric acid-(6’-methylresorufin) ester—DGGR) previously used to assess lipase activity in microplate assays [27,28]. Porcine pancreatic lipase was again used to develop standard curves of activity. Hydrolysis of DGGR results in production of a bluish-purple chromophore (methylresorufin) which was assessed at 580 nm after 30 minutes of incubation with standards/samples and substrate at 37 °C using the same microplate reader described above and following a previously described methodology [28].

### 2.3. Statistical Analysis

Data were analyzed using Prism version 7.04 (GraphPad Software, San Diego, CA, USA). The potential for bias for the two assay methods was assessed by Bland–Altman test. Data were not normally distributed, so non-parametric tests such as the Friedman’s test (to compare paired samples) and the Spearman-rho correlation test (to assess correlations with demographic factors) were used. A *p*-value of less than 0.05 was considered statistically significant.

## 3. Results

### 3.1. Lipolytic Activity

The two assays for lipolytic activities gave divergent results. Most site-specific samples (153 out of 168 or 91%) were positive for lipolytic activity by the olive oil assay, while only 40% (17 out of 42) of whole-saliva samples had detectable lipolytic activity. In comparison, a much lower proportion of site-specific saliva samples (23%, or 39 out of 168) and whole-saliva samples (26%, or 11 out of 42) tested positive by the DGGR method.

A Bland–Altman analysis of the median bias between methods (olive oil assay—DGGR assay) was 15.62 μg/mL (95% limits of agreement −160.0 to 182.6 μg/mL), suggesting that the DGGR method tended to give a lower lipolytic value. A further evaluation of 38 paired assay outcomes were considered where both assays had given a positive result. The median bias from this analysis (16.23 μg/mL, 95% limits of agreement −0.73 to 25.48 μg/mL) showed a similar tendency for the DGGR assay to estimate a lower lipolytic value when activity was positive from both assays. Summary data of the lipolytic activity of site-specific and whole-saliva samples are presented below in Table 1. There was no difference in the lipolytic activity of samples from all sites as assessed by the DGGR assay (Friedman’s test *p*-value = 0.341). The lipolytic activity of samples assessed by the olive oil assay was significantly different (Friedman’s test *p*-value <0.0001), with post-hoc tests noting that saliva sampled from the parotid (median (interquartile range)) of 20.7 (11.7–31.0 μg/mL) and sublingual (18.4 (10.6–47.2 μg/mL) sites had significantly higher activity than that of whole saliva (0.0 (0.0–35.7 μg/mL).

### 3.2. Relationship between Demographic Factors and Lipolytic Activity

Data collected from the lifestyle questionnaire were analyzed using a Spearman-rho test to detect any correlation between the various lifestyle habits and relative lipolytic activity. Parameters where responses were common among all participants in the study (e.g., occupation, age, and health status) were omitted from this analysis. The correlation test results for both assays using olive oil and DGGR substrates are tabulated in Table 2 below. No significant correlations (*p* > 0.05 for all tests) were found between any of the demographic/lifestyle factors and lipolytic activity levels (results from DGGR assay not shown). Estimated frequency of self-reported, habitual high-fat meal consumption ranged from 0 to 20 (median = 2) meals per week and hours spent being physically active from 0 to >5 (median = 2) hours per week within the questionnaire responses. There was no correlation (see Table 2, *p* > 0.05) noted between these lifestyle factors and lipolytic activity assessed by the olive oil assay.

## 4. Discussion

The findings of the current study highlight the presence of lipolytic activity in the oral cavity of some but not all individuals. As activity seems to be higher at the opening of specific saliva glands (the parotid and sublingual sites), this implies that lipolysis is driven by a factor secreted in saliva. To the authors’ knowledge, there is no defined lipase enzyme that has been confirmed to be secreted by human salivary glands [29], although there is some previous evidence to suggest that some lipases can be found in the gustatory tissue of the human tongue that are not analogues of lingual lipase found in other mammalian species [17,30]. The presence of lipolytic activity cannot be considered as absolute evidence of lipase secretion, as it is possible that lipolysis is driven by microbially produced enzymes [31]. There is also the potential that gastric lipase or even pancreatic lipase have reached the oral cavity by gastroesophageal reflux or duodenogastro-oesophageal reflux respectively [32,33] and thus would account for the measured lipolytic activity. Nonetheless, the methodology used here for site-specific sampling helps minimize the potential for interference from other lipase sources.

While both assays suggest low lipolytic activity in some individuals, the proportion of individuals who are “lipase positive” and the absolute level of lipolytic activity by the two assay methods are divergent. The data from pancreatic lipase standards suggest that both methods have a similar minimum detection limit and predictive potential (data not shown). Surface active proteins in saliva have been suggested to interact with the oil:water interface of micelles and reduce droplet size [34], although this action would be expected to increase turbidity at a constant oil content [35]. Extraneous detergents only seem likely to be present in the oral cavity as a result of toothpaste/other hygiene products, although the protocol used here limits the potential for their presence. As the DGGR substrate can only produce color as a result of the cleavage of a lipid ester bond, it seems less likely that this methodology is at risk of “false positive” findings from factors other than lipase.

Another possible reason that the two methods gave different results could be down to the substrate specificity of the lipolytic enzymes present. The fatty acids present on DGGR that will be cleaved from the synthetic chromophore during lipolysis are dodecanoic (lauric–C12:0) acids [36]. DGGR itself tends to form stable microemulsion in aqueous systems [27]. While the specific olive oil used here was not analyzed for its composition, the major fatty acids present tend to be longer and mainly unsaturated, with a large proportion reported to be oleic acid (C18:1) in most oils [37]. Substrate specificity has been noted to occur in other examples of well-characterized lipases [36,38], with older studies suggesting that lingual lipases more rapidly cleave C8:0–C12:0 fatty acids than oleic acid [39]. More recent work also suggested differential impacts of oral processing on free fatty acid release from high-fat foods in the presence and absence of a pancreatic lipase inhibitor [2]. The interfacial action of lipases provides an extra level of complexity in terms of their substrate specificity. Based on the findings here and with the assumption that both assays were able to detect low levels of lipolytic activity equally well in saliva samples, the current findings provide a preliminary suggestion of substrate specificity for lipases found in human saliva. However, it must be noted that only two substrates have been tested within the studies described here, one of which (DGGR is not entirely representative of a dietary glyceride). Additional kinetic studies with a wider range of well-characterized substrates (by fatty acid composition as well as the physical properties of the droplets) would be necessary to help confirm this hypothesis and potentially elucidate further detail of substrate specificity of lipases that appear to be frequently in saliva.

The observation that whole saliva was less frequently positive for lipolytic activity (by both assay methods) than site-specific samples is interesting and, to the authors’ knowledge, not previously reported. The sampling approach is likely to have included secretions from the major salivary glands, which between them produce more than 90% of total saliva [40]. As such, it seems unlikely the higher/more frequently positive lipolytic activity noted in the site-specific samples was a simple result of a dilution factor. This could further suggest that the lipolytic agent is being produced by topical microbes rather than being secreted within the saliva. Previous work has suggested that previously refluxed pepsin may be endocytosed by the mucosa of the upper aerodigestive tract before being subsequently reactivated [41]. It is possible that a similar mechanism could occur with refluxed gastric lipase. It is also possible that intracellular lipases released from sloughed cells [42] may occur at much higher concentrations at the mucosal surface and then either become diluted or inactivated within whole saliva. The data from the two different assays suggest that a proportion of the participants tested did not have detectable lipolytic activity within their oral cavity. Approximately 55% (23 out of 42) of participants provided samples (site-specific and whole saliva) that were all negative for lipase when tested by the DGGR assay, compared to none by the olive oil assay. Other studies that have attempted to assess lipolysis in the human mouth note high degrees of variability, suggesting that some individuals tested may have also exhibited no lipolytic activity [2,17,18].

In terms of overall study design, the authors note that study participants were recruited by convenience sampling around campus which has likely limited the applicability of these findings to a wider population. All participants were of Singaporean nationality and within a tight age range. Further demographic data may have been useful for consideration of differences between different sub-groups. However, such information did not align with the aims of the current project and were not collected to reduce participant burden and improve privacy. It would be expected that a larger sample size would be necessary in future studies to ensure statistical power for such sub-analyses but information such as ethnicity and body weight status may also be important factors to consider [43]. More detailed information of habitual dietary habit, such as that collected from a culturally appropriate food frequency questionnaire would be of value in further research on how diet may correlate with oral lipolytic activity [44]. Microbial metagenomic analysis of site-specific saliva samples would help to define which lipases are being produced by oral microbes in future studies. Simpler approaches such as culturing site-specific oral microbes may also have value but were outside the remit of the current project. While the current study aimed to recruit individuals in generally good health, it was not possible to rule out the potential for gastric reflux contaminating saliva samples. The approach to saliva sampling (particularly rinsing out the mouth before saliva sample was collected) aimed to minimize the potential for such effects. The most accepted methods of gastric assessing gastric reflux currently require highly invasive testing within a clinical setting which would add significant participant burden. Co-detection of other factors from gastric juice (e.g., pepsins) could be considered in future studies, although it is still unclear what amount of pepsin would be considered unphysiological in saliva samples [45].

Proteomic analysis of saliva samples alongside assessment of lipolytic activity seems the most relevant approach to define whether secreted lipases exist. Saliva plays crucial roles in dental health, ease of swallowing, oral taste sensation and therefore potentially dietary preference. The authors believe that the site-specific sampling approach undertaken here is a useful means of evaluating whether saliva outputs from different glands varies across a wide range of parameters, although note that the interpretation of such findings can still potentially be confounded by factors originating from microbes, recent food intake or oral hygiene and sloughed mucosal cells.

## 5. Conclusions

These findings suggest that at least 45% of individuals have lipolytic activity within their oral cavity but at a level that is unlikely to be meaningful outside of taste sensation. These results do not absolutely confirm the presence of a lipase that is secreted within the oral cavity and further work is required to evaluate whether such an enzyme exists in humans.

## Figures and Tables

**Table 1 nutrients-11-00191-t001:** Calculated median (interquartile range) lipolytic activity for saliva samples.

	Lipolytic Activity (Expressed as μg/mL of Pancreatic Lipase Equivalents)				
Assay	Whole saliva	Parotid	Inner lip	Dorsal tongue	Sublingual
DGGR	0.0 (0.0–1.9) ^a^	0.0 (0.0–0.0) ^a^	0.0 (0.0–0.0) ^a^	0.0 (0.0–4.9) ^a^	0.0 (0.0–0.0) ^a^
Olive oil	0.0 (0.0–35.7) ^b^	20.7 (11.7–31.0) ^c^	10.4 (4.2–25.7) ^b,c^	15.4 (8.4–38.5) ^b,c^	18.4 (10.6–47.2) ^c^

DGGR—1,2-o-dilauryl-rac-glycero-3-glutaric acid-(6’-methylresorufin) ester. Saliva samples on same row with different superscripts are statistically different (*p* < 0.05) by Friedman’s test with post-hoc analysis.

**Table 2 nutrients-11-00191-t002:** Summary values from Spearman *r* analysis to assess the correlation between the relative activity and selected lifestyle factors.

	Site of Saliva Sample				
Factor	Whole saliva	Parotid gland	Inner lip	Dorsal tongue	Sublingual
Exercise (hour/week)	0.132 (0.405)	0.097 (0.540)	−0.044 (0.781)	0.228 (0.147)	0.074 (0.642)
High-fat food frequency (meals/week)	−0.145 (0.359)	−0.278 (0.075)	−0.074 (0.642)	−0.012 (0.718)	0.025 (0.873)

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
