# Peer review of "Analysis of the Lipolytic Activity of Whole-Saliva and Site-Specific Secretions from the Oral Cavity of Healthy Adults"

_nutrients, 2019, doi:10.3390/nu11010191_

Round 1
Reviewer 1 Report
Minor grammar corrections required otherwise good manuscript.
Reviewer 2 Report
The article by Yuen et al describes the analysis of lipolytic activity of whole saliva and site specific secretions from the oral cavity of healthy adults. The content that the article tries to address is very interesting since no previous study has reported the secretion of a defined lipase enzyme by the salivary glands in humans. However, a major concern that I have is as follows:
Since the study particularly addresses lipolytic activity, more substrates in addition to olive oil and DGGR should be tested. The authors mentioned about "substrate" specific activity in the Discussion section but testing more substrates would significantly improve the quality of the manuscript and would perhaps allow to draw more definite conclusions.
Minor concern: Please describe how site specific sampling was carried out more clearly to ensure it is clear to the readers that there were minimal chances of contamination from other sites in the oral cavity.
Reviewer 3 Report
This study is concise, conducted in humans with non-invasive procedures, therefore feasible to replicate and extend to a larger population. Methods used seem fine although I am not personally familiar with the analytic methodology used to assessed the lipolytic activity of saliva. The study design is simplistic and with limitations as explained here.
The number of participants is small and very selective as it was extracted from a campus population with a homogeneous age and ethnic population. So the results validity is not yet applicable to a larger and mixed population. The authors recognize the limitations in the discussion. It should be mention in the abstract too.
Based on the information provided by the authors it is an original contribution and is taken as such. It is assumed that the authors know well the available literature. The references provided are to the point and specific to the information provided.
The easy access of the oral cavity and sampling of saliva, facilitate investigations on the possible role of oral microbiota, fat taste and dietary behavior (references 18,19) opening new avenues for better understanding of eating preferences and dietary behavior.
In the discussion reflux is considered as possible source of lipolytic substances, however little information is provided regarding investigation of symptoms associated to reflux within the population included in the study. What about cultures of saliva samples to assess saliva microbiota of the different collection sites? If some of these information could be added will make a better publication. At least an explanation why the information was not collected shall be provided. Any further studies should explore these options in the study design.
Round 2
Reviewer 2 Report
The authors have addressed the concerns raised in the revised version of the manuscript.